# GOAT: efficient and robust identification of gene set enrichment
Frank Koopmans ⬤ ✉

Gene set enrichment analysis is foundational to the interpretation of high throughput biology. Identifying enriched Gene Ontology (GO) terms or disease-associated gene sets within a list of gene effect sizes that represent experimental outcomes is an everyday task in life science that crucially depends on robust and sensitive statistical tools. We here present GOAT, a parameter-free algorithm for gene set enrichment analysis of preranked gene lists. The algorithm can precompute null distributions from standardized gene scores, enabling enrichment testing of the GO database in one second. Validations using synthetic data show that estimated gene set *p*-values are well calibrated under the null hypothesis and invariant to gene list length and gene set size. Application to various real-world proteomics and gene expression studies demonstrates that GOAT identifies more significant GO terms as compared to current methods. GOAT is freely available as an R package and user-friendly online tool for gene set enrichment analyses that includes interactive data visualizations: https://ftwkoopmans.github.io/goat.

High throughput studies of biological systems, using experimental assays such as proteomics and RNA-sequencing, are widely used to identify differentially expressed genes (DEGs) across experimental conditions. To perform systematic and unbiased interpretation of the genes associated with each experimental condition, researchers commonly employ gene set enrichment analysis (also referred to as gene set analysis) to identify which biological processes and (sub)cellular components in the Gene Ontology (GO) database are overrepresented among these DEGs[1–4].

Over-representation analysis (ORA) is the most commonly used type of gene set analysis method and is available through many tools[5–8]. A gene set can be any set of genes of interest; it is typically defined as a set of genes that are known members of the same biological pathway, localized to the same (sub)cellular compartment, co-expressed under certain conditions or associated with some disorder as defined in a gene set database such as GO or KEGG[4,9,10]. The ORA approach tests whether a set of "most important genes" (the foreground set, e.g., significant genes identified in a study) is overrepresented in a gene set (e.g., biological pathway), as compared to a background set of genes (e.g., all genes evaluated in the study). To perform the statistical test, a contingency table is first constructed for each gene set and then a Fisher's exact test or hypergeometric test is applied[4].

While this method has elegant simplicity, is fast to compute and widely available, it has three major disadvantages when used for the interpretation of statistical outcomes from OMICs-based studies; (1) not all available information is used. OMICs-based studies yield gene lists where each gene

has a *p* value and/or effect size that indicates association with the tested phenotype; information on gene rank order and magnitude is lost when partitioning the gene list into the foreground set that contains significant genes and the background set. (2) Two critical fiddle parameters must be specified and these impact ORA results. An arbitrary cutoff for *p* values is required to determine the set of significant genes; which genes are included in the foreground set depends on this *p* value threshold which is typically set to 0.01, 0.05 or the top 10% of smallest *p* values[4,11]. In addition, users must specify the minimum number of significant/foreground genes that must be present in each gene set to include it in ORA. A more stringent setting will reduce the number of evaluated gene sets (with bias) and thereby affect downstream multiple testing corrections. (3) ORA critically depends on using the correct background list but mistakes therein are easily made[12].

The GSEA algorithm[13] is a commonly used permutation-based approach for gene set enrichment analysis that can be applied to a preranked gene list. A preranked gene list is here defined as a table of gene identifiers and their respective effect sizes and/or *p* values that indicate association with some experimental condition (e.g., summary statistics from an OMICS-based study). For each gene set, an enrichment score is calculated based on rank-transformed effect size values from respective genes and this score is compared against a permutation-based empirical null distribution of the same size (i.e., equal number of genes as the target gene set). Analogously, GSEA can also use rank-transformed gene *p* values as input for gene set enrichment testing. An improved implementation of this algorithm

Department of Molecular and Cellular Neurobiology, Center for Neurogenomics and Cognitive Research, Amsterdam Neuroscience, VU University, 1081 HV Amsterdam, The Netherlands.
✉ e-mail: frank.koopmans@vu.nl

is available in the commonly used fGSEA R package[14]. Alternative variants of the GSEA approach are also available in Python packages GSEApy[15] and blitzGSEA[16].

An alternative to GSEA is the recently introduced iDEA algorithm which performs joint modeling of both gene-level and gene set-level enrichment analyses through a hierarchical Bayesian framework[17]. Whereas GSEA operates on either gene $p$ values or effect sizes as input, iDEA takes as input a gene list with both log2 fold changes and their standard errors. In the iDEA paper, authors have shown that this computational framework has more power than GSEA through simulation studies. However, this does come at the cost of orders of magnitude longer computation time as compared to fGSEA.

In this manuscript, we focus on the ubiquitous use case of gene set enrichment analyses where pre-ranked gene lists with $p$ values or effect sizes/foldchanges should be tested for overrepresentation in gene set databases such as the GO database. Alternative methods may be of interest when working specifically on pathway databases with gene–gene causality information[18] (not available in GO) or to identify enriched gene sets directly from gene expression matrices[17,19] or GWAS data[20].

We here present the Gene set Ordinal Association Test (GOAT) algorithm for gene set enrichment testing in preranked gene lists, demonstrate soundness using simulation studies and benchmark the algorithm using various proteomics and gene expression studies. Our results show that GOAT is very fast, testing thousands of GO gene sets completes in 1 s, and identifies more gene sets across six OMICs-based datasets than ORA, GSEA and iDEA. GOAT robustly works with any preranked gene list, from small lists of 100 genes up to 20,000 genes, of any type of data distribution (e.g., provided gene effect sizes do not have to fit some specific distribution and this effect size distribution can have long tails with outliers). Implementations of GOAT are provided as both an R package and user-friendly online tool, making this new approach to highly sensitive gene set enrichment testing available to a wide audience from bioinformatician to biologist.

## Results
### Algorithm overview

We here present GOAT, a novel algorithm for gene set enrichment testing in preranked gene lists that uses bootstrapping to generate competitive null hypotheses (Fig. 1). GOAT uses squared gene rank values (of $p$ values or effect sizes) as gene scores to boost top-ranked genes in the input gene list, which are considered most important when interpreting study outcomes in the biological context. Rank transformation increases the robustness of the user-provided input gene statistics and ensures that downstream gene set scoring and computation of null distributions always operate on a similar gene score distribution, regardless of the input gene list $p$ value/effect size values nor their distribution; hence the GOAT gene score distribution is only variant to the length of the gene list.

Gene set scores are defined as the mean of their respective gene scores. With this approach, gene score distributions are right-skewed (long tail for high gene scores) and so are the empirical gene set null distributions for small gene sets. Following the central limit theorem, this converges to a normal distribution for large gene sets[21] (Fig. 1, Supplementary Fig. 1). For a given gene list length, an empirical null distribution is estimated for each gene set size (number of assigned genes) through extensive bootstrapping procedures and a skew-normal distribution is fitted to each. We provide precomputed null distributions for all gene lists of length 100–20,000 and all gene set sizes, so a typical application of GOAT will skip the generation of null distributions (step 3 in Fig. 1) and thus completes in 1 s for any gene list (Supplementary Fig. 2). The GOAT algorithm has been made available as an open-source R package (c.f. Methods).

### Evaluating gene set $p$ value accuracy under the null hypothesis

We first validated that gene set $p$ values estimated by GOAT are accurate under the null hypothesis, i.e., no surprisingly weak or strong $p$ values should be found when analyzing random gene lists[12,22]. We generated synthetic gene lists of 500, 2000, 6000 and 10,000 genes and applied the GOAT algorithm to test for enrichment across 200,000 randomly generated gene sets of sizes 10, 20, 50, 100, 200 and 1000. Expecting gene set enrichment algorithms to yield uniform gene set $p$ values when testing against many random gene sets, this allowed us to specifically check for potential bias in gene list length or gene set size (e.g., are $p$ values inflated for small gene sets that contain only 10 genes?). Our simulation studies confirmed that gene set $p$ values estimated by GOAT accurately match expected values regardless of gene list length or gene set size (Fig. 2).

Further, we applied similar analyses to the GSEA algorithm, as implemented in the fGSEA R package, and found inaccuracies under default settings that could be rescued by greatly increasing the number of permutations that the algorithm performs (Fig. 2, Supplementary Fig. 3). We, therefore, recommend setting the "nPermSimple" parameter in fGSEA to 50,000 instead of the default 1000, at the minor inconvenience of increasing computation time from the order of seconds to ~1 min when testing thousands of gene sets (Supplementary Fig. 2). Summarizing the differences between observed and expected values using root mean square errors (RMSE) we find that both GOAT and fGSEA (at 50,000 iterations) are well calibrated and consistent across gene list length and gene set size (Fig. 2, Supplementary Fig. 4). When gene lists with $p$ values were used as input, the average RMSE across all simulated gene set sizes and gene list lengths was 0.0045 and 0.0062 for GOAT and GSEA, respectively. Using gene lists with effect sizes as input, prompting GOAT and GSEA to test each gene set for both up- and downregulation (i.e., enrichment in direction of positive/negative effect sizes), the average RMSE was 0.0108 for GOAT and 0.0067 for GSEA.

By repeating these bootstrapping analyses 200,000 times, we were able to accurately assess the calibration of GOAT and fGSEA under the null hypothesis, across various gene set sizes and gene list lengths. We did not extend this analysis to include iDEA because of its computational complexity; analyzing 6000 gene sets with iDEA took ~5 h on a high-

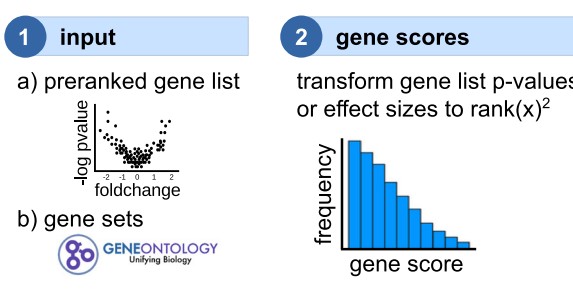

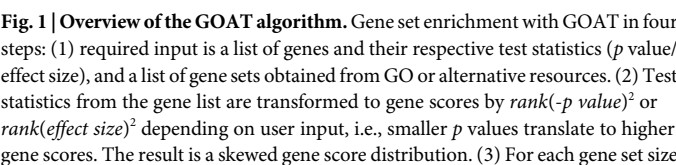

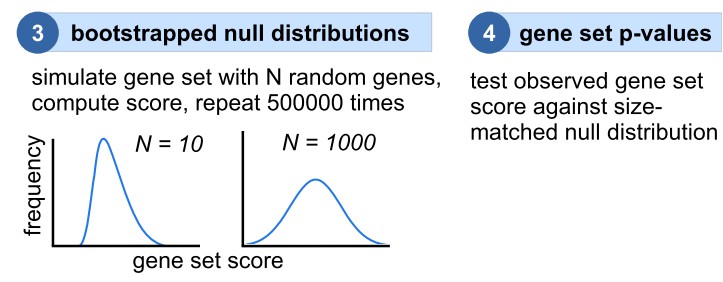

**Fig. 1 | Overview of the GOAT algorithm.** Gene set enrichment with GOAT in four steps: (1) required input is a list of genes and their respective test statistics ($p$ value/effect size), and a list of gene sets obtained from GO or alternative resources. (2) Test statistics from the gene list are transformed to gene scores by $rank(-p\ value)^2$ or $rank(effect\ size)^2$ depending on user input, i.e., smaller $p$ values translate to higher gene scores. The result is a skewed gene score distribution. (3) For each gene set size

$N$ (number of genes), bootstrapping procedures generate a null distribution of gene set scores. This yields a skew-normal distribution for small gene sets and converges to a normal distribution for large gene sets. (4) To determine gene set significance, the score (mean of respective gene scores) is tested against the skew-normal distribution that represents the size-matched null distribution (same gene list length and gene set size).

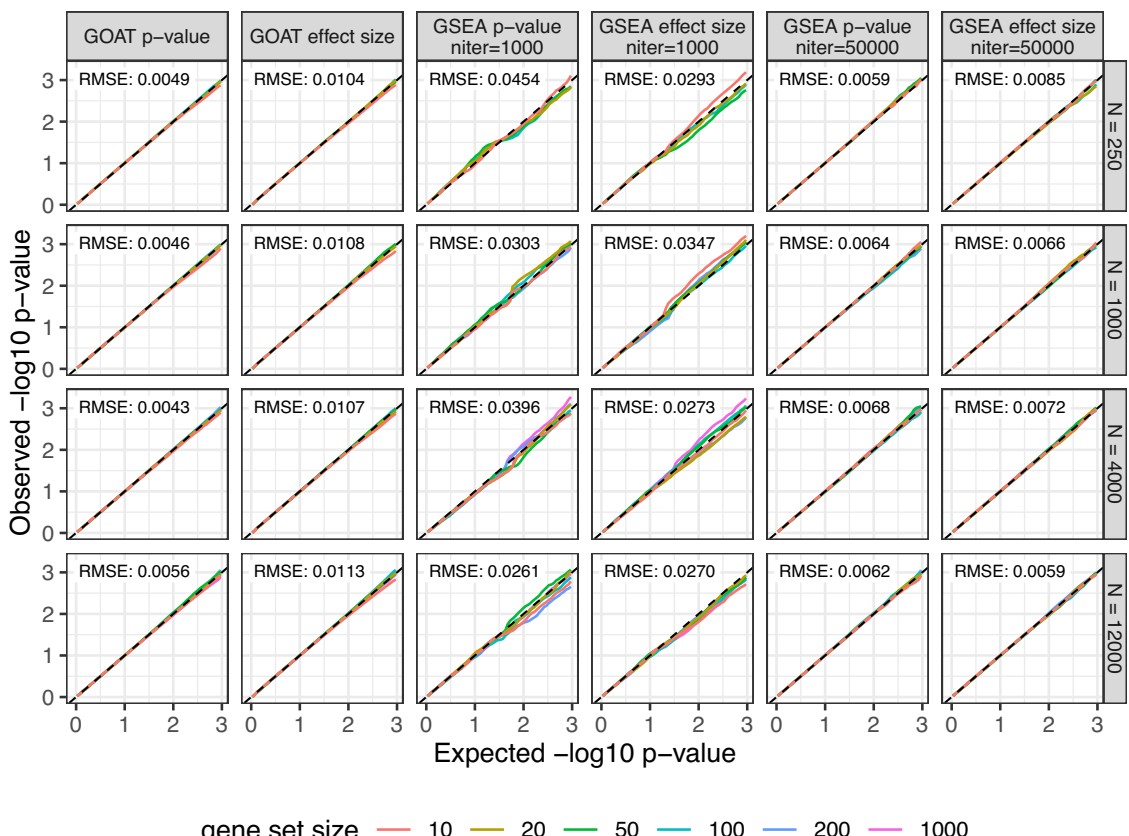

gene set size — 10 — 20 — 50 — 100 — 200 — 1000

**Fig. 2 | GOAT _p_ values are accurate under the null hypothesis in simulations that generate gene lists and gene sets of various sizes.** Random gene lists of 250, 1000, 4000 and 12,000 genes (_N_) were generated to check for potential bias in gene list size (rows). A total of 200,000 synthetic gene sets of 10, 20, 50, 100, 200 or 1000 genes were generated by random sampling of _k_ genes from the respective gene list to check for potential bias in gene set size (line colors). The _y_-axis shows the observed gene set _p_ value for a given _p_ value threshold on the _x_-axis (both on −log10 scale, _p_ values as-is without multiple testing correction). Since randomized gene sets were used in these simulations, expected values are on the diagonal (dashed line). The root mean square error (RMSE) in each panel summarizes the differences between expected and observed −log10 _p_ values across all gene set sizes.

performance workstation computer, extending this to multiple analyses of 200,000 gene sets was not deemed feasible (Supplementary Fig. 2). However, Fig. 2 of the iDEA paper[17] describes an alternative type of null simulations where comparisons between iDEA and fGSEA showed both are reliable but the latter was more accurate.

**Evaluating gene set enrichment sensitivity in simulated data**
We first investigated the empirical gene rank distributions for gene sets that are enriched in the Colameo et al. RNA-sequencing dataset[23] according to the GSEA algorithm (i.e., independent of and unbiased toward GOAT) and found that gene sets with a strong _p_ value typically tend to follow similar patterns of enrichment for top-ranked genes (Fig. 3a). Similarly, gene sets with lowest significance according to GSEA typically exhibit uniform gene rank (position in the input gene list) distributions or slight depletion of top-ranked genes (Fig. 3b). The observed patterns for enriched and background gene sets were comparable across multiple OMICs-based studies (Supplementary Figs. 7–10). Informed by these empirical data, we setup a series of simulation studies to compare the sensitivity of gene set enrichment algorithms in a controlled environment with expected outcomes by generating synthetic gene sets with various degrees of enrichment.

Two classes of gene sets were generated: null gene sets containing 100 genes that were uniformly sampled from the input gene list and "foreground" gene sets that contain 100 genes that are drawn from the input gene list in a pattern that matches the enriched gene sets observed in the empirical data (Fig. 3a–c). We thus expect that upon gene set enrichment testing, the former group yields gene set _p_ values that are higher than the latter group. And indeed, receiver operating characteristic (ROC) analyses showed that

GOAT can differentiate between 10,000 randomly generated null gene sets (uniformly sampled, no enrichment of top-ranked genes) and 1000 randomly generated gene sets that follow a gene sampling pattern of enrichment for top-ranked genes (Fig. 3d). Where GOAT achieved perfect separation with 5% partial Area Under Curve (pAUC) at 95% specificity, GSEA achieved similar results at 4.77% pAUC. In contrast, the application of iDEA with default settings resulted in only 0.23% pAUC. Next, we generated foreground gene sets with a weaker enrichment pattern and found that GOAT still achieved a 5% pAUC whereas GSEA performance dropped to 4.14% (Fig. 3e, f). In order to check if any of the evaluated methods generated false positives in the ROC analyses we generated foreground gene sets using the same uniform sampling pattern as the null gene sets, thus expecting no separation between foreground and null genets, and found that all methods returned expected results (Fig. 3g, h).

Whereas GOAT and GSEA apply rank transformation to input data, iDEA does not and this might lead to a lack of robustness when dealing with input gene lists with unexpected input data distributions. For example, gene list log2fc distributions might have a wide variety of shapes beyond a standard normal (not perfectly centered, heavy skew, long tails, etc.) and thus not suit model assumptions in the current version of the iDEA hierarchical Bayesian model. We observed heterogeneity among iDEA input data distributions across OMICs-based datasets (Supplementary Fig. 5) and thus implemented two types of data preprocessing for iDEA; in analyses denoted as iDEA* we rescaled the estimated gene log2fc variation (an iDEA parameter) such that the distribution over all genes in the input gene list approximates a gamma distributions with shape 2 and scale 0.5. In analyses denoted as iDEAb* we rescaled the input gene log2fc distributions by their

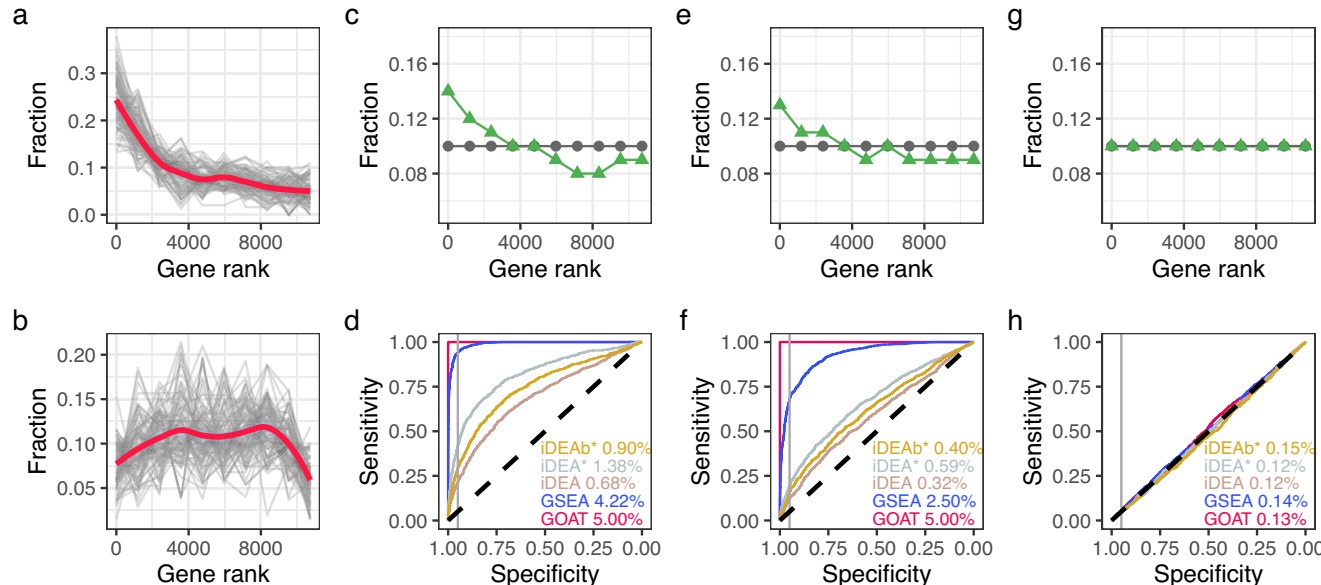

**Fig. 3 | GOAT demonstrates superior sensitivity in application to synthetic datasets.** Gene rank distribution for constituents from the top (**a**) and bottom (**b**) 10% most significant downregulated GO terms (containing 50–500 genes) in the Colameo et al. RNA-sequencing dataset, as detected by GSEA. **c, e, g** Templates used to simulate 10,000 null gene sets (gray dots) and 1000 foreground gene sets (green triangles). Values on the y-axis represent the proportion of random genes drawn from a respective bin in the input gene list (x-axis) to generate a gene set of 100 genes. **d, f, h** Gene set enrichment analyses were applied to synthetic gene sets in the preceding figure panel. The ROC curves reflect the rate at which respective methods assign a smaller p value to foreground gene sets, enriched for top-ranked genes, as compared to null gene sets that were drawn from uniform distributions. **g, h** As a control, foreground gene sets were also drawn from a uniform distribution. Inset values for each method represent the partial Area Under Curve (pAUC) at 95% specificity (vertical gray line). iDEA*: iDEA with rescaled beta_var, iDEAb*: iDEA alternative model (testing beta only) with rescaled beta (c.f. Methods).

median absolute deviation and then apply iDEA with "modelVariant" set to TRUE such that iDEA directly models the (now standardized) log2fc values. As a result, we found that the iDEA analyses with rescaled input data yielded better ROC performance than the default model (Fig. 3d, f).

Clear separation between foreground and null gene sets was observed for GOAT across multiple patterns of enrichment, from easier templates where foreground gene sets contained relatively many top-ranking genes to templates where foreground gene sets contained fewer top-ranking genes, as observed through p value distributions (Supplementary Fig. 6) and ROC from simulations across four datasets (Supplementary Figs. 7–10). Throughout all analyses, GOAT outperformed GSEA and both are more sensitive and consistent as compared to iDEA.

### Setup for application to real-world datasets

After confirming that gene set p values estimated by GOAT are accurate, we applied our algorithm to various real-world datasets (c.f. Methods) to compare its sensitivity to GSEA (fGSEA implementation), iDEA and the classical hypergeometric test (ORA). Using a gene list with effect sizes as input allows GOAT and GSEA to perform a two-sided evaluation of enrichment for either gene up- or downregulation. In contrast, using a gene list with (only) p values will perform a one-sided test of enrichment per gene set (there is no information on up/downregulation) and this may lead to a strong reduction in significant gene sets as also observed in previous studies[24]. Note that iDEA only works with one specific input data format, log2 foldchanges and their respective standard errors (which we derived from gene p values in all benchmark datasets in accordance with the iDEA manual), so iDEA can also perform a two-sided test for gene set up- and downregulation. Multiple testing correction was independently applied per gene set "source" (i.e., GO MF/CC/BP) using Bonferroni adjustment at $\alpha = 0.05$ and we subsequently adjusted all p values to account for these three separate tests across GO domains. To allow for fair comparison between gene set enrichment methods we consistently applied the exact same multiple testing correction procedures.

### GOAT is more sensitive than alternative methods

Compared to GSEA with gene effect sizes as input, GOAT identified significantly more GO terms across all datasets we evaluated (Fig. 4). The increase across all GO domains was 85% in the Colameo et al. mass spectrometry dataset and 36.4% in the RNA-sequencing dataset[23], 61.5% in the Higginbotham et al. mass spectrometry dataset[25], 335% in the Hondius et al. mass spectrometry dataset[26], 23.4% in the Sahadevan et al. RNA-sequencing dataset[27] and 93.7% in the Wingo et al. mass spectrometry dataset[28].

Similarly, when using gene p values as input the GOAT algorithm outperformed GSEA by 36~418% except for the Higginbotham et al. dataset where GSEA and GOAT identified four and one significant gene sets in total, respectively. The increased sensitivity when using gene effect sizes, as compared to p values as input, suggests that enrichment in evaluated gene sets was in either the up- or downregulated genes. Losing out on this information in a unidirectional test (i.e., using gene p values as input) resulted in fewer significant GO terms with a severity that varied between datasets as has been observed previously[24].

Whereas the ratio of enriched gene set counts between GSEA and GOAT was quite consistent across datasets, iDEA performance varied a lot. Compared to iDEA, GOAT with effect sizes as input identified a similar number of gene sets in the Wingo et al. dataset (185 vs 184) and the Higginbotham et al. dataset (57 vs 63). But whereas iDEA found zero significant gene sets in the Hondius et al. dataset, GOAT identified 74. Similarly, in the Sahadevan et al. dataset iDEA identified 82 significant gene sets while GSEA identified 488 and GOAT 602.

We proceeded to investigate the correlations between the gene set p values estimated by iDEA, GSEA and GOAT and found that iDEA does not produce p values for some gene sets, with a varying rate of missingness (0.4~62%) among datasets even though the input data format and code pipeline was exactly the same throughout and valid log2fc and standard error input values were provided to iDEA in all benchmarks (Supplementary Figs. 11–13). Rescaling input data prior to iDEA application alleviated the rate of missingness, resulting in 0~4.3% for iDEA* and 0.1~19% for iDEAb* (Supplementary Fig. 11, c.f. Methods). However, iDEA with

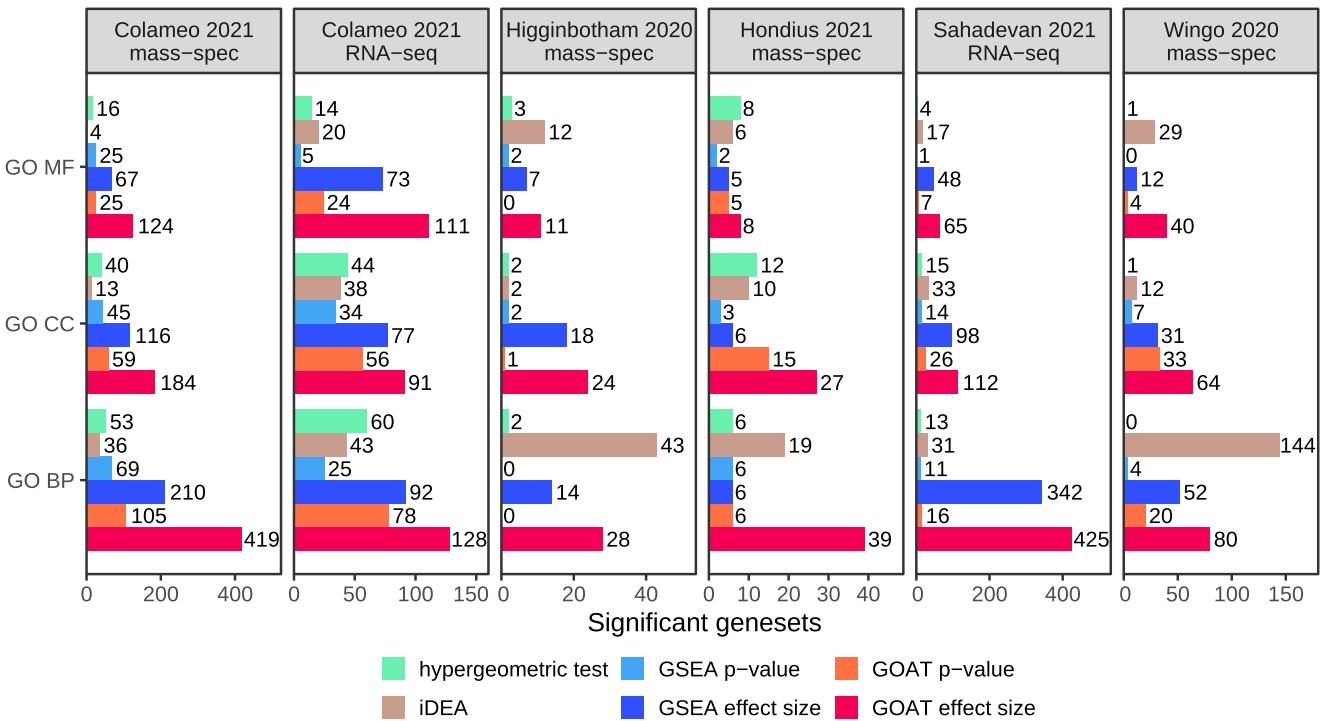

**Fig. 4 | GOAT identifies more significant GO terms compared to GSEA and ORA across six OMICs-based studies.** Differential expression analysis results from each study (panels) were subjected to six methods (colors) to identify enriched GO terms. GO MF, CC and BP represent the respective gene ontology domains molecular functions, cellular components and biological processes. The x-axis shows the number of significant gene sets identified by each method after Bonferroni adjustment at $\alpha = 0.05$ and adjustment for testing three sources of gene sets.

rescaled inputs did not result in improved performance when benchmarking real-world datasets so while our results suggest that a potential cause of the iDEA errors is due to (unknown) expectations for input data distributions our presented concepts offer no structural improvement to iDEA (Supplementary Fig. 14). Future research into the iDEA method could explore whether further improvements to input data preprocessing beyond proof-of-concepts introduced here can consistently abrogate iDEA errors while estimating gene set $p$ values and potentially improve iDEA performance in ROC and real-world benchmarking. Note that this is a non-issue for GSEA and GOAT, which stabilize input gene scores by rank transformation and should therefore be invariant to input data distribution characteristics (i.e., only the rank order matters).

We found no clear trend in the iDEA gene set missingness in terms of expected significance that was estimated via GOAT, i.e., the reduced number of iDEA significant gene sets observed in Fig. 4 for some datasets is not because specifically top-hits are missing in iDEA output (Supplementary Figs. 12, 13). Next, we proceeded to compare iDEA estimated gene set $p$ values to the established GSEA method but found that results were not correlated (Supplementary Fig. 13a). The $R^2$ of gene sets that are among top 25% in either iDEA or GSEA was between 0 and 0.16 across all evaluated real-world datasets. However, a similar comparison between GSEA and GOAT showed a strong agreement of both methods for gene sets with a strong $p$ value. With $R^2$ values of 0.41, 0.68, 0.70, 0.84, 0.87 and 0.90 across datasets we found similar trends of gene set enrichment between GSEA and GOAT, albeit with a generally lower $p$ values for GOAT which suggest a gain in power for GOAT (Supplementary Fig. 13b). Enrichment testing with iDEA took 43 min for the Higginbotham et al. dataset (gene list of 2715 genes) up to 4 h and 42 min for the Sahadevan et al. dataset (13,849 genes) on a high-performance workstation computer.

Comparing classic ORA against GOAT with gene effect sizes as input, GOAT identified 180~9100% more significant gene sets. Comparing classic ORA against GOAT with gene $p$ values as input for a level playing field (same input information), GOAT identified 34~2750% more significant

gene sets except for the Higginbotham et al. dataset where ORA and GOAT identified 7 and 1 significant gene sets in total, respectively, and the Hondius et al. dataset where GOAT and ORA both identified 26 significant gene sets. Although it should be noted that the datasets where ORA performed relatively well as compared to GOAT and GSEA were all relatively small datasets. The Hondius and Higginbotham datasets are relatively small gene lists with 2261 and 2715 genes of which 8% and 7% are "significant", respectively. A potential explanation for the increased number of significant gene sets for ORA in these cases could be due to ORA-specific fiddle parameters; when only retaining gene sets with three significant genes in a short gene list with few significant hits there might be a strong selection bias toward gene sets with some enrichment (as opposed to e.g., a gene list of 10,000 genes with 20% significant proteins), compounded by a reduced multiple testing correction burden.

When comparing gene sets identified by GOAT, using a gene list with $p$ values as input for fair comparison against ORA, i.e., not making use of the directionality of regulation that is encoded in the gene effect sizes, we find the same trend of enrichment between ORA, GSEA and GOAT but with generally stronger $p$ values for GOAT which strengthens our confidence that the GOAT approach does not yield completely unrelated gene sets but instead exhibits both correlation to existing methods and increased power (Supplementary Fig. 15).

## GOAT online
In addition to the GOAT R package, we developed an online tool that makes it easy to apply GOAT to any gene list and subsequently use a rich interactive toolset to generate diverse data visualizations (Fig. 5). The GOAT online tool is entirely implemented in client-side Javascript which enables this tool to perform all computations locally (i.e., in the web browser). This is enabled by the unique design of the GOAT algorithm, where we can test gene set enrichment using precomputed null distributions which thus boils down to computing gene set scores (the mean of respective gene scores) and performing a skew-normal enrichment test. In contrast, porting fGSEA or

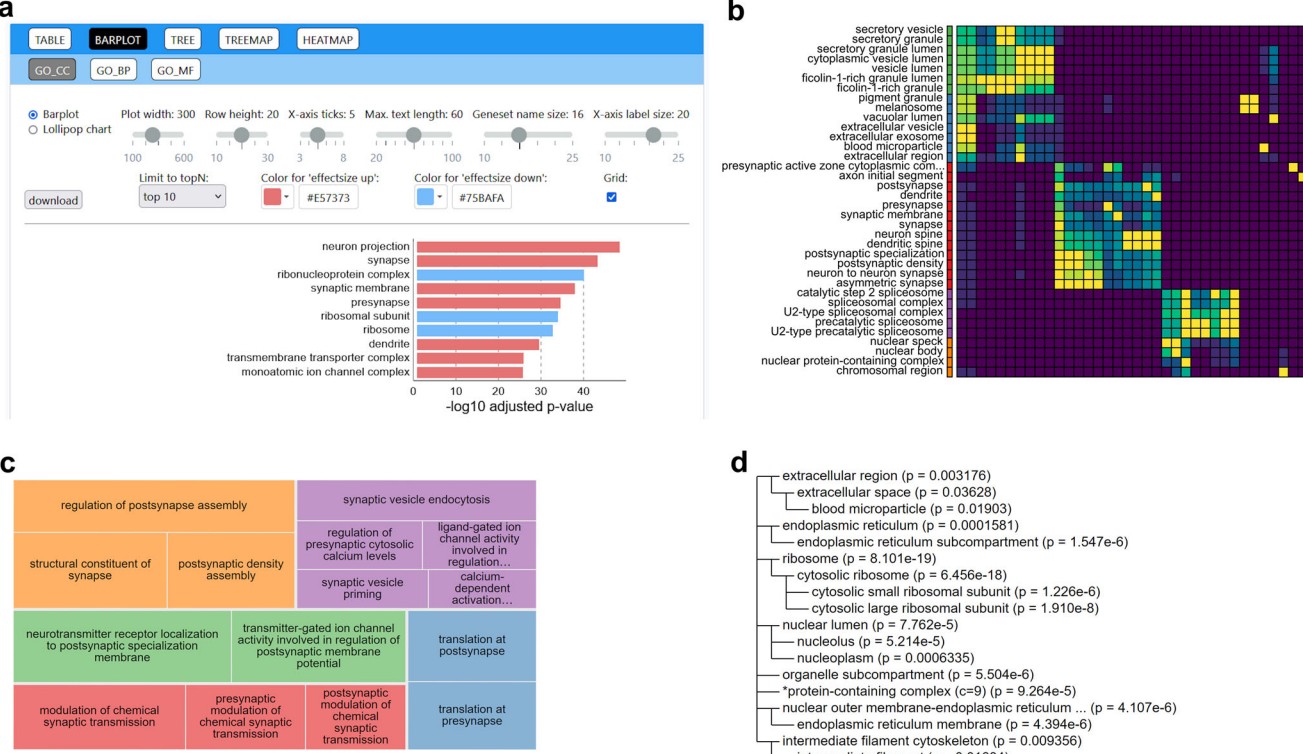

**Fig. 5 | GOAT online visualizations of enriched gene sets. a** Screenshot of the GOAT online tools for creating barplots. Users can interactively customize plot dimensions, label size, pick desired colors and change the bars to "lollipop chart style". Changing any settings will immediately update the figure shown on screen. **b** Example heatmap figure that visualizes the overlap in genes (weighted by gene scores) between each pair of gene sets. This may be used to cluster gene sets based on similarity of annotated genes. Users may customize color coding, clustering parameters, label sizes, etc. Example visualizations of GO parent-term relations among significant gene sets. A treemap (**c**) can be used to reduce identified gene sets into color-coded groups representing high-level ontology terms and inner tiles as respective child terms, with customizations analogous to other figures (plot and label sizes, tile ratios, etc.). Alternatively, the parent relations for all significant gene sets can be visualized as a compact tree representation (**d**) that users may collapse at any level by clicking on a gene set name.

iDEA to a web application would take considerably more effort (and is probably not computationally feasible for the latter). Consequentially, hosting this "static website" does not require special webservers and operation of the website can trivially scale to a large user base.

Gene sets from the GO and SynGO databases are built-in and can be immediately used, but users can also upload their own gene set collection (e.g., KEGG) in the common GMT format. Downloading the result table with all tested gene sets and their GOAT estimated *p* values includes a log file with methods text that is updated for current analyses (i.e., M&M text that reflects settings in GOAT online, GOAT version, GO/SynGO version and citations depending on configured database, etc.). Data visualizations such as barplots, treemaps and heatmaps can be customized interactively. For example, users can change the plot dimensions, text size, color schemes, etc., and the figure will be live-updated. Generated figures can be downloaded as high-quality vector graphics (SVG format). In summary, the GOAT online tool brings the novel GOAT algorithm to the masses and delivers a user-friendly interactive interface to generate publication-ready gene set analyses.

## Discussion

GOAT is a novel algorithm for gene set enrichment analysis of preranked gene lists that is widely applicable to studies that use the GO database, or alternative gene set resources such as SynGO or KEGG, to identify enriched gene sets. The algorithm generates competitive null hypotheses with low computational complexity. Computing the empirical null distributions completes in the order of seconds and when using the precomputed null distributions (default), computation of gene set enrichment *p* values takes a second. Robustness and accuracy have been demonstrated throughout extensive simulation studies that confirmed estimated *p* values are accurate under the null hypothesis and invariant to gene list

length and gene set size. Application of GOAT to six diverse OMICs-based datasets revealed a significant increase in sensitivity as compared to popular gene set enrichment algorithms GSEA and the classical hyper-geometric test (ORA). To encourage the adoption of this novel algorithm, GOAT has been made available as an R package at https://github.com/ftwkoopmans/goat and a user-friendly online tool at https://ftwkoopmans.github.io/goat. The interactive GOAT online webtool can be used to identify enriched gene sets with GOAT and create customized publication-ready data visualizations.

The vast majority of GO analysis tools currently use the ORA approach to testing gene set enrichment, an approach that has been shown to have inferior sensitivity and requires thoughtful configuration of arbitrary fiddle parameters. A recent study scrutinized 186 publications and found that most published functional enrichment studies (that use ORA) suffered from one or more major flaws[12]. In contrast, the GOAT algorithm introduced here is parameter-free and avoids all presented pitfalls. To avoid common shortcomings in the documentation of gene set enrichment procedures identified by Wijesooriya et al., GOAT results include a ready-made methods text that is tailored to user-defined configurations (e.g., selected GO database version).

While gene sampling-based algorithms for gene set enrichment analysis are not new[4], GOAT takes a next step by combining: (1) a gene score metric with a non-linear increase in importance for top-ranking genes, (2) efficient and accurate modeling of null distributions as skew-normal (instead of relying on nonparametric tests) and (3) a caching strategy that allows us to precompute the null distributions (the part of the algorithm with highest computational complexity), thereby reducing the application of GOAT to a simple test that completes in 1 s even when a large gene list of 10,000 genes is tested against the GO database.

The amount of skew in the null distributions monotonically decreases from small to large gene sets, as does their estimated standard deviation. This property is currently used by GOAT to stabilize the empirical null distributions derived from a large number of permutations (500,000) by smoothing estimated skew-normal parameters Sigma and Xi over a range of gene set sizes. Future work could explore the theoretical underpinnings of the GOAT algorithm such that the null distribution of a given gene set size and gene list length might be calculated instead of estimated from bootstrapping. Additionally, future work on GOAT could explore further improvements to the accuracy of estimated gene set $p$ values by eliminating sampling variation (albeit minor at 500,000 iterations) and avoid inaccuracies introduced by fitting skew-normal distributions to empirical null distributions.

The sensitivity of GOAT was first evaluated in a series of simulations where estimated gene set $p$ values were compared between synthetic gene sets with a priori known enrichment and random gene sets without any enrichment of top-ranked genes. ROC analyses revealed GOAT outperformed GSEA and iDEA across all simulated data. Application to six diverse real-world proteomics and gene expression studies demonstrated that GOAT consistently identified more significant GO terms as compared to both GSEA and ORA, with 23–335% improvement per dataset over the popular fGSEA tool (which in turn is an improvement over the widely used original GSEA tool[13,14]) and orders of magnitude over ORA (Fig. 4). The iDEA method was recently proposed as a challenger to GSEA and in our evaluations where the exact same input data and multiple testing adjustments were applied for all methods throughout applications to a diverse set of real-world datasets we found that iDEA was an improvement over GSEA in some, but not all, datasets while GOAT yielded most gene sets overall. Further, we observed that iDEA could not estimate gene set $p$ values at varying rates across simulated and real-world datasets. We hypothesize that the effect size distributions of some datasets used in our studies violate iDEA model assumptions, thus causing problems for iDEA gene set testing, and assume these edge cases can be corrected in future updates to iDEA (Supplementary Figs. 5–14). Taken together with a computation time in the order of hours, compared to minutes for GSEA and 1 s for GOAT, the iDEA method might be more suited for use cases similar to those described in the original manuscript, i.e., same type of input data and preprocessing, but does not seem suitable as a general-purpose solution for gene set enrichment (in its current form) like GSEA and GOAT which are parameter-free and proven to be more robust across simulations and real-world data applications presented here.

With the increased sensitivity of the GOAT algorithm, users will uncover more significant gene sets as shown here in application to various OMICs-based datasets and subsequently, there might be more partially overlapping gene sets among the gene set enrichment results. Future research might focus on improvement to the downstream interpretation of GOAT results, for instance, gene set summarizing tools that combine gene sets with ontological similarity or high gene constituent overlap[29–31]. To take a first step we have implemented gene set clustering tools in the GOAT R package and GOAT online webtools.

Taken together, the GOAT algorithm is fast, portable, parameter-free, robust and more sensitive than existing methods for gene set analysis. Two implementations of GOAT are made freely available; an R package for use in bioinformatic workflows and a user-friendly online tool.

## Methods
### GOAT implementation
We have implemented the GOAT algorithm in an open-source R package that is available at https://github.com/ftwkoopmans/goat. Version 1.0 was used for all analyses presented here. Gene sets can be automatically downloaded from the GO database using the Bioconductor[32] infrastructure. Alternatively, GO gene sets can be imported directly from "gene2go" files that are available through NCBI so users can control which version of the GO annotations they use and reproduce their analyses at any time using the exact same gene set definitions as input (independent of Bioconductor versions). To work with alternative gene set definitions beyond GO, users may import gene sets from the synaptic gene knowledgebase SynGO[33] or

import gene sets in the commonly used GMT format. Crucial parts of the (bootstrapping part of the) GOAT algorithm were implemented in optimized C++ code to enable efficient generation of permutation-based null distributions. As a result, finding enriched GO terms with the full GOAT algorithm using 500,000 bootstrap iterations completes in the order of seconds on a typical personal computer (Supplementary Fig. 2). Using the precomputed null distributions, GOAT completes in a second.

Multiple testing correction in the GOAT R package and GOAT online is implemented as a two-step process: first, we apply Bonferroni adjustment independently applied per gene set "source" (i.e., GO MF/CC/BP). However, users may also opt for the Benjamini–Hochberg procedure. Second, Bonferroni adjustment is applied according to the number of gene set sources that were tested. For example, when working with the GO database a $p$ value correction (in addition to the first step) is applied to account for three separate that are applied tests across GO domains MF, CC and BP. In Fig. 4, significant gene sets at Bonferroni adjustment at $\alpha = 0.05$ are shown.

### Implementation of alternative gene set enrichment methods
Comparative analyses with ORA (i.e., Fisher's exact or hypergeometric test), GSEA (using the fGSEA R package) and iDEA described in this manuscript were also performed using the GOAT R package, in which we integrated all algorithms to facilitate benchmarking using the same codebase and workflow (e.g., same filtering/selection for input gene sets and same multiple testing correction procedures). fGSEA version 1.22.0 and iDEA version 1.0.1 were used for all analyses presented here.

We implemented various workflows for iDEA; the default workflow (referred to as "iDEA") applies the iDEA R package with default settings and "louis $p$ value calibration" method per the iDEA manual. Alternatively, we implemented a rescaled variant of the default iDEA workflow where the estimated gene log2fc variations (an iDEA parameter) are rescaled such that the distribution over all genes in the input gene list approximates a gamma distribution with shape 2 and scale 0.5, which is referred to as "iDEA*". In addition, we added the option to rescale input gene log2 foldchanges by their overall median absolute deviation (a simple but robust approach to standardize) and then apply iDEA per the manual with "modelVariant = TRUE" such that iDEA directly models the (now standardized) log2fc values which is referred to as "iDEAb*". For all iDEA-based results presented here, "Louis $p$ value calibration" was applied unless noted otherwise (e.g., in Supplementary Fig. 6).

### Statistics and reproducibility
The GOAT algorithm is introduced in the "Results" section (Fig. 1), with the full implementation available in the open-source GOAT R package. The GitHub repository at https://github.com/ftwkoopmans/goat provides all R code (version 1.0 specifically is available at https://github.com/ftwkoopmans/goat/tree/6ea11e25a22fb509ee9ac624efc1fad7c668ba49[34]), including functions to apply GOAT and alternative gene set enrichment algorithms (see documentation in file "test_genesets.R"), and scripts that generate the figures presented here ("analyses" directory in the GitHub repository). All source data that is required to reproduce our analyses are provided here as Supplementary Data.

### Datasets and databases
For the Colameo proteomics and gene expression data[23], we used the results from the "Compartment" test available in supplementary data and defined the foreground genes for ORA as proteins with adjusted $p$ value of at most $10^{-4}$ on account of the large number of significant hits even at this stringent threshold. The proteomics dataset yielded a gene list with 3935 genes, the gene expression gene list contained 11,922 genes. For all other studies, we used the same threshold for adjusted $p$ values as reported in the original study when performing ORA, which was 5% FDR in all cases. The Sahadevan gene expression was imported from Table S8 (condition "24 h") which resulted in a gene list of 13,849 genes[27]. The Higginbotham proteomics data were imported from Table S2 which resulted in a gene list of 2715 genes[25]. The Hondius proteomics dataset was imported from Table S8

(condition "Control vs Tangle") which resulted in a gene list of 2261 genes[26]. The Wingo proteomics data were imported from Table S4 which resulted in a gene list of 8170 genes[28].

The reported proteins/genes in each study were mapped to NCBI Entrez human gene identifiers using the GOAT R package. For records that mapped to the same gene (e.g., protein isoforms) we retained the single entry with strongest $p$ value per gene.

All datasets are available through the GOAT R package for the convenience of repeating presented analyses without having to download the source data and repeat data preparations; after loading the goat R package, issue the R statement *download_goat_manuscript_data()*. The R code that was used to process these data tables into the gene lists used in this manuscript is provided at the GOAT GitHub repository in the "analyses" directory, together with all code that generated the figures presented in this manuscript.

GO data from NCBI gene2go, downloaded at 2024-01-01, were loaded into the GOAT using the *load_genesets_go_fromfile()* function and this version of the GO database was used for all GO analyses. For SynGO, release 1.2 was obtained from https://www.syngoportal.org and gene sets were loaded into GOAT using the *load_genesets_syngo()* function.

## Data availability
All datasets analyzed in this study have been previously published and were obtained from respective Supplementary Data files. The preprocessed data are available as an RData file at the GOAT GitHub repository at https://github.com/ftwkoopmans/goat/ (c.f. Methods). The source data behind the graphs in the paper can be found in Supplementary Data 1.

## Code availability
The GOAT R package and all code that was used to perform the presented data analyses are available at https://github.com/ftwkoopmans/goat/ [34].

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

## Competing interests

The authors declare no competing interests.
