## [Peer Review File · Communications Biology]

Reviewers' comments:

Reviewer #1 (Remarks to the Author):

Koopmans presents a new approach to functional class scoring based enrichment analysis called GOAT. I appreciate that the design of GOAT is parameter-free, as novices using alternative tools stumble into problems with how to select genes to be part of the background (ORA) and how to rank genes (GSEA). Having reasonable defaults is important to avoid such pitfalls. I tested the software (R and web) using the example data and it worked as advertised. I think this software is an important advance and this manuscript is worthy of publication. I have only a few points to address.

Recommendation: Minor revision.

1. Line 43: Mention Tarca as a reference for the top % of genes (PMID: 24260172).
2. Line 151: Please check the previous/closed issues for this package for a solution (<https://github.com/xzhoulab/iDEA/issues>), and if this issue is new, please open a new issue, I would prefer iDEA maintainers provide an explanation before publication.
3. The software should report the enrichment score, as this is a proxy for effect size in gene set tests.
4. The software package should be deposited to bioconductor or CRAN.

Optional

5. The discussion describes "comprehensive simulation", but the simulation work shows FDR control only, not the degree of sensitivity/recall. Two approaches to this: (1) start with a real dataset with a large sample number per group (n), and downsampling it, comparing GOAT and other approaches. And (2) the use of synthetic changes to expression levels to genes of interest and comparing GOAT with other approaches. I appreciate the amount of work for these is considerable, which is why it is indicated as optional, but it would strengthen the argument of using GOAT over competitors like GSEA.

Reviewer #2 (Remarks to the Author):

Brief summary: GOAT is a new tool designed to evaluate the enrichment of gene sets such as those available in the Gene Ontology database. Though many gene set enrichment analysis tools are already available, GOAT uses pre-calculated, estimated null distributions to reduce computation time. GOAT deviates from the Subramanian et al. algorithm for gene set enrichment analysis because it transforms input gene scores, scores gene sets using the mean transformed gene scores, uses estimated null distributions, does not provide enrichment scores, and adjusts p-values instead of calculating the FDR based on the distribution of normalized enrichment scores.

Overall impression: GOAT performs gene set enrichment analysis faster than the Subramanian et al. GSEA algorithm by using pre-calculated, estimated null distributions instead of empirical distributions which account for each given input distribution and also compromising on outputs like enrichment scores. While the results suggest that GOAT can identify more gene sets as significant, additional analysis is needed to assess potential false discovery (e.g., using validation cases) and thus better compare GOAT to other gene set enrichment analysis tools.

Specific comments:

1. I do not see an explanation of the value of using GOAT over other gene set enrichment analysis tools in the introduction. I think it would be beneficial to add this information to the introduction since

there are many other gene set enrichment analysis tools.

2. Can a performance metric for GOAT be calculated based on the simulation study? For example, what fraction of replicates resulted in inflated p-values for each combination of gene set sizes and gene list lengths?

3. The results indicate that GOAT identifies more gene sets as significant compared to GSEA and iDEA. However, it is not described how this supports the claim that GOAT is more sensitive than these methods. It would be beneficial to test these algorithms on cases where specific gene sets are expected to be positively or negatively enriched (using both simulated data and independently published data sets) and also compare GOAT to weighted GSEA which considers input gene scores instead of just gene ranks.

4. The colors in the legend of Figure S6 do not match those in the figure itself. I suggest using shapes to distinguish the methods illustrated in Figure S6 to accommodate any colorblind readers.

Reviewer #3 (Remarks to the Author):

Major comments on the paper:

Identifying "more" significant terms does not mean identifying the correct terms. The authors need to come up with a better benchmark that shows that their method identified the "correct" terms better than other methods.

The authors compared their method to ORA, and iDEA, and GSEA but there are other methods?

Minor comments on the paper:

Geneset, genelist, and effectsize should be separate words.

What is the difference between gene list and gene set?

What is a foreground set?

There are several other fast GSEA packages and algorithms.

Comments on the website:

What does the GOAT acronym stand for?

The site says genelist but the paper geneset. Which one is it?

More GMT files should be made available.

What is SynGO?

Heatmap spills over

Visualizations are not intuitive what is effect size up and effect size down?

What is M&M text?

Should be able to perform the analysis for GO and SynGo all at once.

Reviewer #1 (Remarks to the Author):

Koopmans presents a new approach to functional class scoring based enrichment analysis called GOAT. I appreciate that the design of GOAT is parameter-free, as novices using alternative tools stumble into problems with how to select genes to be part of the background (ORA) and how to rank genes (GSEA). Having reasonable defaults is important to avoid such pitfalls. I tested the software (R and web) using the example data and it worked as advertised. I think this software is an important advance and this manuscript is worthy of publication. I have only a few points to address.

Recommendation: Minor revision.

1. Line 43: Mention Tarca as a reference for the top % of genes (PMID: 24260172).

Thanks for this suggestion, I have added the reference to line 43.

2. Line 151: Please check the previous/closed issues for this package for a solution (<https://github.com/xzhoulab/iDEA/issues>), and if this issue is new, please open a new issue, I would prefer iDEA maintainers provide an explanation before publication.

I agree and have thoroughly investigated the matter. My apologies for the long-winded reply to this question, I here provide a lot of background information that does not really have a place in the manuscript but hopefully helps address your concerns.

There is no solution to the problems I encountered with iDEA in previous GitHub issues. Similar problems have been posted before (e.g. <https://github.com/xzhoulab/iDEA/issues/20> or <https://github.com/xzhoulab/iDEA/issues/25>) but it seems like all GitHub reports of similar errors are edge-cases where users for example do not provide the complete input gene list, which does not reflect the application of iDEA in the work presented here (same input for iDEA as for GOAT and GSEA; always the entire gene list) but does suggest this error is related to unexpected/incompatible input data. Further, it seems like most examples and GitHub tickets make use of similar input data (scRNA-seq data, e.g. processed with Seurat).

Prior to submitting the GOAT manuscript I already contacted the iDEA authors via email on January 29th to describe the issues and included a minimal example with code&data to reproduce the problems, but unfortunately received no response (<https://xiangzhou.github.io/software/> Shiquan Sun is listed as contact, I mailed to sqsunsph@xjtu.edu.cn as instructed). Perhaps I should have emailed all authors instead or just make a GitHub ticket back in January. However, I wanted to move this issue forward so in the meanwhile I have added a number of additional analyses specifically for iDEA, encouraged by your comments, that shine a light on the errors generated by iDEA.

I have created an R script that applies both iDEA and fgSEA to an example dataset and observes many errors for iDEA but not for fgSEA. This minimalistic script without further dependencies can be found on the GOAT GitHub site at;

https://github.com/ftwkoopmans/goat/blob/main/analyses/compare_idea_fgsea_standalone-script.R

The previous version of the GOAT manuscript stated on lines 155-158: *“Future research into the iDEA method could explore if these issues relate to the gene list log₂fc distributions that are used as input, which might have a wide variety of shapes beyond a standard normal (not perfectly centered, heavy skew, long tails, etcetera) that might not suit model assumptions in the current iDEA hierarchical Bayesian model.”*

New analyses seem to confirm the hypothesis that observed iDEA errors are at least in part related to the input gene list data distributions. The manuscript has been updated with visualizations of gene list data distributions from various real-world datasets (highlighting stark differences in data distributions in Figure S5), two proposed solutions for rescaling input data prior to iDEA and evaluation thereof in the real-world dataset benchmarks and newly added ROC simulations (Figures 3, S6-S11, S14). In short, new figures show that rescaling input data prior to application of iDEA strongly reduces the error rate (i.e. proportion of gene sets for which it fails to estimate p-values) but unfortunately did not elevate iDEA's performance when benchmarking real-world datasets (i.e. proposed rescaling did not structurally improve upon iDEA).

Important differences between GOAT/fgSEA and iDEA are that the latter has stricter input filtering by default regarding gene set size and the number of differentially expressed genes per gene set. From the tutorial (<https://xzhoulab.github.io/iDEA/>) we learn that the “min_precent_annot=0.0025” parameter restricts gene sets to some proportion of the gene list length so a gene list of e.g. 8000 genes will be tested only for gene sets that contain at least 20 genes. The “min_degene” parameter, which defaults to 5, is described as *“the threshold for the number of detected DE genes in summary statistics. For some of extremely cases, the method does not work stably when the number of detected DE genes is 0”*. Because we want to make a fair comparison between GOAT, fgSEA and iDEA we apply each method on the exact same gene list and collection of gene sets, then apply the same multiple testing correction to each method. This might be an additional cause of iDEA errors (besides the rescaling of input data distributions); our workflow for fair method comparison will put some gene sets into iDEA that might be filtered out by iDEA under default parameters.

Note that GOAT and fgSEA also work reliably across a huge range of gene list lengths (Figure 2). However, iDEA requires large gene list to sample from as suggested by iDEA authors at GitHub ticket <https://github.com/xzhoulab/iDEA/issues/25> ; a gene list of “only 2091” genes might be a problem whereas GOAT and fgSEA can be applied even when e.g. users wants to test an immunoprecipitation-based proteomics dataset that contains 100~200 genes (not a problem in the analyses presented in the paper where all gene lists contain thousands of genes, but does imply that GOAT and fgSEA are more general purpose).

Further, iDEA does return a competitive number of gene sets for some gene lists but lags behind for some other gene lists (Figure 4). This shows that there is not a structural problem with the implementation of iDEA throughout presented benchmarks; every analyses uses the exact same GOAT R function `test_gene_sets_idea()`.

To conclude, in the discussion section I do not dismiss this method as inferior altogether but instead frame this as a tool that works well with certain datasets, whereas GOAT and fgSEA are faster (thus more practical) and robust to any type of input data (thus general purpose). This is described in the discussion section, lines 328-340:

The iDEA method was recently proposed as a challenger to GSEA and in our evaluations where the exact same input data and multiple testing adjustments were applied for all methods throughout applications to a diverse set of real-world datasets we found that iDEA was an improvement over GSEA in some, but not all, datasets while GOAT yielded most gene sets overall. Further, we observed that iDEA could not estimate gene set p-values at varying rates across simulated and real-world datasets. We hypothesize that the effect size distributions of some datasets used in our studies violate iDEA model assumptions, thus causing problems for iDEA gene set testing, and assume these edge-cases can be corrected in future updates to iDEA (Figures S5-S14). Taken together with a computation time in the order of hours, compared to minutes for GSEA and one second for GOAT, the iDEA method might be more suited for use-cases similar to those described in the original manuscript, i.e. same type of input data and preprocessing, but does not seem suitable as a general-purpose solution for gene set enrichment (in its current form) like GSEA and GOAT which are parameter free and proven to be more robust across simulations and real-world data applications presented here.

3. The software should report the enrichment score, as this is a proxy for effect size in gene set tests.

The raw gene set scores computed by the GOAT algorithm are relative to their respective null distribution that is gene set size-matched, so this metric is easily misinterpreted downstream because these scores are not comparable between gene set sizes. I agree that it is useful to provide some standardized enrichment score and not only gene set p-values; the updated R package now includes standardized z-scores with the GOAT gene set testing results.

4. The software package should be deposited to bioconductor or CRAN.

The R package has been updated such that it is fully compatible with CRAN. It was tested on Windows/Linux/MacOS. I have submitted the package to CRAN and it passed all the automated validations. However, I'm currently awaiting (human in the loop) review of the package by CRAN and this might take some time. But since I spent all allotted rebuttal time on the new data analyses and R package updates, I decided not halt the review process until the GOAT package is live on CRAN and have submitted this rebuttal to Nature Communications. Once the R package passed CRAN review, I will push the exact same version (1.0) and updated documentation to the GOAT GitHub repository as well (and create a Zenodo archive thereof, to be linked in the manuscript).

I hope this is satisfactory and you can greenlight this topic provided that the package is live on CRAN at the end of the review process, of course.

Optional

5. The discussion describes “comprehensive simulation”, but the simulation work shows FDR control only, not the degree of sensitivity/recall. Two approaches to this: (1) start with a real dataset with a large sample number per group (n), and downsampling it, comparing GOAT and other approaches. And (2) the use of synthetic changes to expression levels to genes of interest and comparing GOAT with other approaches. I appreciate the amount of work for these is considerable, which is why it is indicated as optional, but it would strengthen the argument of using GOAT over competitors like GSEA.

Thanks for these ideas, I have followed up on this topic with an extensive set of new analyses inspired by your second approach.

The “comprehensive simulations” quote from the submitted manuscript referred to the simulations that are used to validate the GOAT gene set p-value accuracy under the null hypothesis, which goes further than other manuscripts that evaluate gene set enrichment methods; the GOAT manuscript explores both gene list length and gene set size as potential confounders, which I have not seen in any other manuscript, and performs 200k permutations to increase estimation confidence (accuracy of Figure 2).

I agree with your suggestion that more simulations strengthen the case for GOAT; the updated manuscript includes additional simulations to explore sensitivity/recall. The new Figures 3 and S6-S10 show how synthetic gene sets that should not be enriched (uniformly sampled gene constituents) *versus* additional gene sets with artificial enrichment of genes with a strong effect size (should have better p-value than uniform sampling) perform in ROC analyses. Analyses include 4 distinct patterns of synthetic gene set enrichments, e.g. from “easy to distinguish from background” to difficult and a baseline test. GOAT outperforms alternative methods throughout.

In summary: the new analyses show, under various data simulation scenarios, that gene set p-values estimated by GOAT are better separated between *a priori* known true- and false-positives as compared to competing methods. Results are described in the section “Evaluating gene set enrichment sensitivity in simulated data”, starting on line 133.

Reviewer #2 (Remarks to the Author):

Brief summary: GOAT is a new tool designed to evaluate the enrichment of gene sets such as those available in the Gene Ontology database. Though many gene set enrichment analysis tools are already available, GOAT uses pre-calculated, estimated null distributions to reduce computation time. GOAT deviates from the Subramanian et al. algorithm for gene set enrichment analysis because it transforms input gene scores, scores gene sets using the mean transformed gene scores, uses estimated null distributions, does not provide enrichment scores, and adjusts p-values instead of calculating the FDR based on the distribution of normalized enrichment scores.

Overall impression: GOAT performs gene set enrichment analysis faster than the Subramanian et al. GSEA algorithm by using pre-calculated, estimated null distributions instead of empirical distributions which account for each given input distribution and also compromising on outputs like enrichment scores. While the results suggest that GOAT can identify more gene sets as significant, additional analysis is needed to assess potential false discovery (e.g., using validation cases) and thus better compare GOAT to other gene set enrichment analysis tools.

Specific comments:

1. I do not see an explanation of the value of using GOAT over other gene set enrichment analysis tools in the introduction. I think it would be beneficial to add this information to the introduction since there are many other gene set enrichment analysis tools.

Thanks for pointing out this oversight. I have summarized four main strengths of GOAT at the end of the introduction (fast, robust, more significant gene sets, easy to use). Lines 72-81:

“We here present the GOAT algorithm for gene set enrichment testing in preranked gene lists, demonstrate soundness using simulation studies and benchmark the algorithm using various proteomics and gene expression studies. Our results show that GOAT is very fast, testing thousands of GO gene sets completes in 1 second, and identifies more gene sets across 6 OMICs-based datasets than ORA, GSEA, and iDEA. GOAT robustly works with any preranked gene list, from small lists of 100 genes up to 20 thousand genes, of any type of data distribution (e.g. provided gene effect sizes do not have to fit some specific distribution and this effect size distribution can have long tails with outliers). Implementations of GOAT are provided as both an R package and user-friendly online tool, making this new approach to highly sensitive gene set enrichment testing available to a wide audience from bioinformatician to biologist.”

2. Can a performance metric for GOAT be calculated based on the simulation study? For example, what fraction of replicates resulted in inflated p-values for each combination of gene set sizes and gene list lengths?

Great idea, I agree that having a summary score in addition to the figure/visualization allows for a better comparison especially when differences are small and therefore difficult to see in the plots. The manuscript now includes the RMSE for each figure panel in the null simulations (Figure 2, S3). Further, I have added a new Figure S4 that shows all RMSE values from main Figure 2 together to allow for easy comparison across methods and gene set sizes. This is described in the result section, lines 118-124.

3. The results indicate that GOAT identifies more gene sets as significant compared to GSEA and iDEA. However, it is not described how this supports the claim that GOAT is more sensitive than these methods. It would be beneficial to test these algorithms on cases where specific gene sets are expected to be positively or negatively enriched (using both simulated data and independently published data sets) and also compare GOAT to weighted GSEA which considers input gene scores instead of just gene ranks.

Thanks for the input and ideas, I have followed up with an extensive set of new analyses.

The idea was that I showed that 1) both GSEA and GOAT have well calibrated p-values and 2) GOAT consistently identified more significant gene sets in real-world data, and thus concluded that GOAT is more sensitive because the increased number of hits in application to 6 real-world datasets was not at the cost of an inflated false positive rate (as demonstrated by null simulations in Figure 2).

However, I agree that additional analyses would strengthen this claim. Inspired by your comments and those of other reviewers, the updated manuscript includes new ROC analyses that explore the sensitivity of GOAT under various data simulations. The new Figures 3 and S6-S10 show how synthetic gene sets that should not be enriched (uniformly sampled gene constituents) *versus* additional gene sets with artificial enrichment of genes with a strong effect size (should have better p-value than uniform sampling) perform in ROC analyses. Analyses include 4 distinct patterns of synthetic gene set enrichments, e.g. from “easy to distinguish from background” to difficult and a baseline test. GOAT outperforms alternative methods throughout.

In summary: the new analyses show, under various data simulation scenarios, that gene set p-values estimated by GOAT are better separated between *a priori* known true- and false-positives as compared to competing methods. Results are described in the section “Evaluating gene set enrichment sensitivity in simulated data”, starting on line 133.

> "also compare GOAT to weighted GSEA which considers input gene scores instead of just gene ranks"

I checked the fGSEA R package's manual and found that the "gseaParam" can be used to change the weight of gene ranks, but found no option to use raw gene scores (e.g. log2fc values as-is) with fGSEA. Searching the online documentation, I found a relevant comment by the fGSEA author at <https://github.com/ctlab/fgsea/issues/45> which suggests that the 4 options in the original GSEA tool by Broad institute (classic, weighted, weighted_p2, weighted_p3) correspond to fGSEA option "gseaParam" being equal to 0, 1, 2, 3, respectively (where 1 is the default fGSEA setting). So all GSEA results in the GOAT manuscript, and any previously published use of fGSEA, make use of "gseaParam=1" which seems to correspond to "weighted GSEA".

I proceeded to test if "gseaParam=2" leads to improvement when applying fGSEA to real-world datasets, but found it decreased the number of significant gene sets identified in every evaluated dataset (Figure S14, as compared to Figure 4).

4. The colors in the legend of Figure S6 do not match those in the figure itself. I suggest using shapes to distinguish the methods illustrated in Figure S6 to accommodate any colorblind readers.

Thanks for spotting this, the color alpha was previously not in sync in the figure legend. I have updated this figure (now labeled Figure S15) to amend the discrepancy in colors and used shapes as suggested.

Reviewer #3 (Remarks to the Author):

Major comments on the paper:

1. Identifying “more” significant terms does not mean identifying the correct terms. The authors need to come up with a better benchmark that shows that their method identified the “correct” terms better than other methods.

Thanks for this comment, I have worked on an extensive set of new analyses to address this and believe your suggested analyses have led to an improved manuscript.

The idea was that I showed that 1) both GSEA and GOAT have well calibrated p-values and 2) GOAT consistently identified more significant gene sets in real-world data, and thus concluded that GOAT is more sensitive because the increased number of hits in application to 6 real-world datasets was not at the cost of an inflated false positive rate (there is none in the null simulations).

However, I agree that additional analyses would strengthen this claim. Inspired by your comments and those of other reviewers, the updated manuscript includes new ROC analyses that explore the sensitivity of GOAT under various data simulations. The new Figures 3 and S6-S10 show how synthetic gene sets that should not be enriched (uniformly sampled gene constituents) versus additional gene sets with artificial enrichment of genes with a strong effect size (should have better p-value than uniform sampling) perform in ROC analyses. Analyses include 4 distinct patterns of synthetic gene set enrichments, e.g. from “easy to distinguish from background” to difficult and a baseline test. GOAT outperforms alternative methods throughout.

In summary: the new analyses show, under various data simulation scenarios, that gene set p-values estimated by GOAT are better separated between *a priori* known true- and false-positives as compared to competing methods. Results are described in the section “Evaluating gene set enrichment sensitivity in simulated data”, starting on line 133.

2. The authors compared their method to ORA, and iDEA, and GSEA but there are other methods?

Indeed, the manuscript only compares GOAT against commonly used (ORA, (f)GSEA) and recent methods (fGSEA, iDEA). In the introduction, we outline the scope of the manuscript on line 67:

In this manuscript we focus on the ubiquitous use case of gene set enrichment analyses where preranked gene lists with p-values or effect sizes/foldchanges should be tested for overrepresentation in gene set databases such as the Gene Ontology database. Alternative methods may be of interest when working specifically on pathway databases with gene-gene causality information¹⁸ (not available in GO) or to identify enriched gene sets directly from gene expression matrices^{17,19} or GWAS data²⁰.

Thus we do not benchmark/compare against tools that for example perform gene set enrichment testing directly from input gene expression data matrices. The goal of current data analyses is to compare GOAT against alternative methods that also operate on “preranked gene lists”, i.e. algorithms that solve the same problem for their user (given the exact same input data).

The fGSEA method is a clear improvement upon the original GSEA approach introduced by Subramanian et al., it works very well and represents the state-of-the-art for the GSEA approach; we can assume that alternative implementations of the same algorithm will not greatly outperform fGSEA in terms of p-value accuracy (e.g. null simulations shown in Figure 2) or sensitivity (e.g. simulations in Figure 3 or application to real-world datasets in Figure 4).

This is also apparent from alternative implementations of the GSEA algorithm by Subramanian et al. in blitzGSEA and GSEApY (now cited in the GOAT manuscript on line 60), where authors mostly describe Python availability of GSEA by virtue of their new tools and report these are quite fast in computing results, but do not report any gains in the number of identified/significant gene sets in practical use. Because the same gene set testing approach is applied in these tools I do not expect radical improvement over fGSEA in terms of their p-value accuracy or sensitivity, especially given that we run fGSEA with 50000 permutations which will be very hard to beat with any implementation/variant/approximation of the GSEA algorithm.

On this account I would like to add a technical note; it seems that the blitzGSEA manuscript compared against fGSEA with default settings for the latter. This can be seen in the blitzGSEA paper’s code that is available at https://github.com/MaayanLab/blitzgsea/blob/main/testing/fgsea_benchmark.ipynb (scan for lines that read “fgseaRes <- fgsea(“ to find their usage of fGSEA); it does not specify the “nPermSimple” parameter. However, the GOAT manuscript shows that increasing the number of iterations greatly improves fGSEA accuracy (Figure 2, S3). The default value for this parameter is 1000 but in the GOAT R package we always change this to 50000 to optimize GSEA results. Thus, fGSEA results shown in the GOAT manuscript should represent best-in-class results for any GSEA implementation.

To the best of my knowledge, fGSEA and iDEA are state-of-the-art for gene set enrichment analysis of preranked gene lists (scope of this manuscript). For example, the iDEA manuscript already showed in Figures 2-4 that CAMERA (Wu et al. 2012) and PAGE (Kim et al. 2005) are clearly outperformed by both fGSEA and iDEA. I agree that if there is a gene set enrichment method that operates on preranked gene lists (scope of this manuscript) that has been shown to be accurate (i.e. no inflated false positive rate) and more sensitive than GSEA (i.e. significantly “better” than (f)GSEA), it would be interesting to see how it compares to GOAT.

Minor comments on the paper:

3. Geneset, genelist, and effectsize should be separate words.

Thanks for pointing this out. I have updated the manuscript, supplementary information and labels throughout all figures accordingly.

4. What is the difference between gene list and gene set?

I agree that it is good to clarify this up-front and have updated the introduction section. Line 30 now introduces the definition of gene sets in accordance to Maleki et al. (with reference);

A gene set can be any set of genes of interest; it is typically defined as a set of genes that are known members of the same biological pathway, localized to the same (sub)cellular compartment, co-expressed under certain conditions or associated with some disorder as defined in a gene set database such as GO or KEGG^{4,9,10}.

Line 52 now clearly defines gene lists;

A preranked gene list is here defined as a table of gene identifiers and their respective effect sizes and/or p-values that indicate association with some experimental condition (e.g. summary statistics from an OMICS-based study).

5. What is a foreground set?

Line 33 introduces the ORA approach and defines “foreground set”;

The ORA approach tests whether a set of “most important genes” (the foreground set, e.g. significant genes identified in a study) is overrepresented in a gene set (e.g. biological pathway), as compared to a background set of genes (e.g. all genes evaluated in the study).

6. There are several other fast GSEA packages and algorithms.

I agree, it is useful to cite alternative GSEA implementations and added references to recent publications of the GSEAPy and blitzGSEA tools on line 60. Please refer to my comments on your second question for further discussion on alternative GSEA tools.

Comments on the website:

Thanks for all the extensive feedback and attention to detail, this is very helpful. I have added more documentation throughout the website and believe this has improved its usability. Please find my answers to your questions below.

7. What does the GOAT acronym stand for?

The GOAT acronym is now defined at the start of the landing page and in the documentation section.

8. The site says gene list but the paper gene set. Which one is it?

I have added the same definitions for gene list and gene set that are used in the updated manuscript to the documentation section of the website.

9. More GMT files should be made available.

I have elaborated the “custom gene set database” section of the documentation. The MSigDB website that is described there is a great resource for non-GO gene set databases. Instead of mirroring (and keeping in sync with) its data I prefer to advertise their resource in the GOAT online documentation and encourage GOAT online users to browse the MSigDB website for databases that might be appropriate for them. The updated documentation should help users navigate the MSigDB website and retrieve GMT files of choice, then use these with GOAT online.

10. What is SynGO?

The documentation section now features a glossary that includes GO, SynGO, KEGG and references to respective websites. A short description of SynGO has also been added to the GOAT online tool.

11. Heatmap spills over

The heatmap is intended to scale to any arbitrary size; users can rescale the figure as desired (with the “Cell size” slider) to either a small plot or a large figure to scrutinize details. Allowing scaling to large sizes is also valuable because it is a live preview of the SVG file (same size/scale) that is obtained when downloading the figure.

12. Visualizations are not intuitive what is effect size up and effect size down?

I have added an explanation to the user interface for all data visualizations (i.e. BARPLOT / TREE / TREEMAP / HEATMAP). Specifically per your question about the barplots, the mouse over the help symbol for “effect size down” now shows the help text;

“Color for gene sets that are enriched in gene constituents that have a negative effect size in your dataset / input gene list”

Similar clarifications have been added throughout the data visualizations.

13. What is M&M text?

The updated website no longer contains this out of context shorthand. The explanation of result files in the “Result summary” has been updated and includes a reference to the documentation section (where I have added a detailed description of the GOAT online output files).

14. Should be able to perform the analysis for GO and SynGo all at once.

The analysis of each gene set collection/database is currently considered an independent analysis. Unfortunately it is not trivial to rework the user interface and data processing functions to streamline the analyses of multiple collections/databases in one go. I have added your suggestion to the new feature list for future GOAT online updates.

REVIEWERS' COMMENTS:

Reviewer #1 (Remarks to the Author):

I am satisfied that all my points are addressed.

Reviewer #2 (Remarks to the Author):

All of my comments have been adequately addressed.

Reviewer #3 (Remarks to the Author):

The authors addressed all my suggestions and concerns satisfactorily